# Spatial Transcriptomics Reveals Signatures of Histopathological Changes in Muscular Sarcoidosis

**DOI:** 10.3390/cells12232747

**Published:** 2023-11-30

**Authors:** Hippolyte Lequain, Cyril Dégletagne, Nathalie Streichenberger, Julie Valantin, Thomas Simonet, Laurent Schaeffer, Pascal Sève, Pascal Leblanc

**Affiliations:** 1Département de Médecine Interne, Hôpital de la Croix-Rousse, Hospices Civils de Lyon, 69004 Lyon, France; hippolyte.lequain@chu-lyon.fr; 2Institut NeuroMyoGène INMG-PGNM, Physiopathologie et Génétique du Neurone et du Muscle, UMR5261, Inserm U1315, Faculté de Médecine Rockefeller, Université Claude Bernard UCBL-Lyon 1, 69008 Lyon, France; nathalie.streichenberger@chu-lyon.fr (N.S.); thomas.simonet@univ-lyon1.fr (T.S.); 3CRCL Core Facilities, Centre de Recherche en Cancérologie de Lyon (CRCL) INSERM U1052-CNRS UMR5286, Université de Lyon, Université Claude Bernard Lyon1, Centre Léon Bérard, 69008 Lyon, France; cyril.degletagne@lyon.unicancer.fr (C.D.); julie.valantin@lyon.unicancer.fr (J.V.); 4Service d’Anatomopathologie, Centre de Biologie et Pathologie Est (CBPE), Hospices Civils de Lyon, 69500 Bron, France; 5Centre de Biotechnologie Cellulaire, CHU de Lyon—HCL Groupement Est, 69677 Bron, France; 6Pôle IMER, HESPER EA 7425, 69002 Lyon, France

**Keywords:** muscular sarcoidosis, granuloma, skeletal muscle, fibrosis, Visium, spatial transcriptomic

## Abstract

Sarcoidosis is a multisystemic disease characterized by non-caseating granuloma infiltrating various organs. The form with symptomatic muscular involvement is called muscular sarcoidosis. The impact of immune cells composing the granuloma on the skeletal muscle is misunderstood. Here, we investigated the granuloma–skeletal muscle interactions through spatial transcriptomics on two patients affected by muscular sarcoidosis. Five major transcriptomic clusters corresponding to perigranuloma, granuloma, and three successive muscle tissue areas (proximal, intermediate, and distal) around the granuloma were identified. Analyses revealed upregulated pathways in the granuloma corresponding to the activation of T-lymphocytes and monocytes/macrophages cytokines, the upregulation of extracellular matrix signatures, and the induction of the TGF-β signaling in the perigranuloma. A comparison between the proximal and distal muscles to the granuloma revealed an inverse correlation between the distance to the granuloma and the upregulation of cellular response to interferon-γ/α, TNF-α, IL-1,4,6, fibroblast proliferation, epithelial to mesenchymal cell transition, and the downregulation of muscle gene expression. These data shed light on the intercommunications between granulomas and the muscle tissue and provide pathophysiological mechanisms by showing that granuloma immune cells have a direct impact on proximal muscle tissue by promoting its progressive replacement by fibrosis via the expression of pro-inflammatory and profibrosing signatures. These data could possibly explain the evolution towards a state of disability for some patients.

## 1. Introduction

Sarcoidosis is a complex inflammatory disease of unknown etiology characterized by the presence of non-caseating granulomas (NCGs) infiltrating various organs including the lungs and heart [1]. NCGs are dynamic structures composed of fused CD68-positive macrophages surrounded by polarized Th1 and Th17 lymphocytes [2]. Sarcoidosis is considered a multisystemic disease that results from an inappropriate immune response to certain environmental triggers, which activate and stimulate the proliferation of macrophages and T-cells [3,4]. These triggers can be of bacterial origin, as it has been previously described for granuloma formation in patients infected with *Mycobacterium Tuberculosis* [5], *Propionibacterium acnes* [6] or iatrogenic, through aluminum hydroxide crystals present in adjuvant vaccines and previously involved in macrophagic myofasciitis lesions [7,8].

NCGs reported in 50 to 80% of patient’s muscle biopsies that are suspected as being of active sarcoidosis do not exhibit clinical muscle symptoms [9]. When muscle symptoms are present, it is referred to as muscular sarcoidosis, a rare idiopathic inflammatory myopathy occurring in less than 3% of sarcoidosis cases. Diagnosis can be challenging due to the absence of well-defined criteria, so clinical investigations take into account muscular symptoms, as well as biological and imaging findings, alongside the presence of intramuscular NCGs in biopsies [10].

Muscular sarcoidosis can have a severe course in a significant proportion of patients, leading to corticosteroid dependence and disability [11,12]. There are no clear treatment recommendations for this disease, and the therapeutic strategies developed in the literature are based on small retrospective series, given the rarity of the disease. Currently, the classic treatment is corticosteroid therapy alone as a first line with the addition of immunosuppressive treatments in case of relapse [9,11,12].

Unraveling the mechanisms underlying this disease could probably help us in developing appropriate therapeutic strategies.

With the emergence of spatial transcriptomics approaches, it becomes possible to visualize inter-cellular connection circuits and apprehended more complex multicellular communication networks. Single nucleus RNA sequencing, although very informative, may not provide the spatial context needed for this investigation.

Skeletal muscles can be exposed to severe damage that requires rapid and efficient repair. In such situations, adult muscle stem cells (satellite cells) are activated to regenerate damaged muscle fibers. During regeneration, satellite cells are assisted by helper cells, including mesenchymal cells (fibro/adipogenic precursor cells) as well as immune cells, especially macrophages that support and regulate the myogenesis process [13,14]. In muscle disorders like Duchenne muscular dystrophy (DMD), the regenerative process is altered and the muscle tissue is progressively replaced by a fibrotic tissue that prevents regeneration [15].

The direct impact of granuloma formation on skeletal muscle tissue and the interactions between the thousands of cells composing the granuloma and the secretion of immune mediators on the adjacent skeletal muscle tissue remains largely unexplored.

Here, intercellular communications were investigated at the transcriptomic level using a powerful next-generation molecular profiling approach called Visium gene expression (10x Genomics), allowing the spatial visualization of gene expression (about 19,000 mRNA targets) within tissue organization at a 55 μm resolution.

We found that the immune cells composing the granulomas have a direct impact on the proximal muscle tissue by promoting the muscle tissue replacement through fibrosis via the expression of pro-inflammatory and pro-fibrotic cytokines and concomitantly reducing the expression of muscle-specific genes.

## 2. Materials and Methods

### 2.1. Human Samples

Two patients (Patient #1 and Patient #2) were identified for granulomatous myositis associated with multisystemic sarcoidosis and were profiled using spatial transcriptomic technique. They were screened through the MYOLYON database (Hospices civils de Lyon HCL). Muscular sarcoidosis was validated through detection of NCG on muscular biopsies associated with clinical, biological and imaging in favor of muscle involvement and at least another site involved (clinically, imaging or biologically), in accordance with ATS/ERS/WASOG statement on sarcoidosis [16]. A PET scan revealed mediastinum and muscular involvement and a quadriceps biopsy showed muscular granuloma infiltratation in the two patients.

### 2.2. Standard Protocol Approvals, Registrations and Patients Consents

The patients gave their written and informed consent for complementary research on their biopsies, associated with consent to disclose regarding pictures. Human biological samples (biopsies 18EH16366 et 208H22524) and associated data were obtained from Tissu-Tumorothèque Est and Centre de Biologie Cellulaire (CBC) Biotec Biobank (Centre de Ressources Biologiques Hospices Civils de Lyon NF S 96900 certification BB-0033-0046), a center for biological resources authorized by the French Ministry of Research (AC-2019-3465). The study was conducted in accordance with national legislations and the ethical standards laid down in the 1964 Declaration of Helsinki and its later amendments. In accordance with French law, this study was approved by the Institutional Review Board of the Hospices Civils de Lyon (IRB 00013204, request 22_812) and by the national data protection commission (Commission Nationale de l’Informatique et des Libertés; request n° 22_5812).

### 2.3. Antibodies

The anti-CD68 (M0755; dilution IHC 1:50) was from Dako (Santa Clara, CA, USA), the anti-CD3 (2GV6; IHC 1:1) and anti-CD8 (SP57; IHC 1:1) from Ventana (Roche Diagnostics Basel, Switzerland), the anti-HLA-DRA (AB_2848953; dilution IHC 1:100) from Invitrogen, and the anti-CD163 (10D6; IHC 1:200) from Thermo Fisher Scientific (Waltham, MA, USA).

### 2.4. Histology and Immunohistology

Biopsies were performed in quadriceps with local anesthesia for all patients by Dr Nathalie Streichenberger at the anatomopathological department of Pierre Wertheimer Hospices Civils de Lyon. Fresh muscle specimens were divided into two fragments; one was fixed in 10% neutral buffered formalin and embedded in paraffin (Formalin-Fixed Paraffin-Embedded—FFPE sections), and the other frozen in isopentane. Three and five μm thick longitudinal and transversal sections were cut either with a microtome or a cryostat for morphological analysis (Leica instruments GmbH, Hubloch, Hamburg, Germany) and transferred to Superfrost Plus slides (VWR International bvba, Leuven, Belgium) for morphological and immunohistochemistry (IHC) staining. Hematoxylin–phloxine–saffron (HPS) staining was used for morphological analysis both in fixed and frozen sections. An automated IHC analysis of the sections was performed on a BenchMark XT (Ventana Medical Systems Inc., Tucson, AZ, USA) using primary antibodies and revealed via avidin–biotin–peroxidase complex and Ventana DAB detection and amplification kits. Sections were counterstained with hematoxylin and eosin. Slides were scanned using a Zeiss Axio Observer Z1 microscope (Oberkochen, Germany).

### 2.5. Spatial Transcriptomics Using Visium

All the steps were followed according to the 10x Genomics (Pleasanton, CA, USA) protocols (CG000407_VisiumSpatialGeneExpressionforFFPE_UserGuide_RevC.pdf, accessed on 7 April 2022).

The RNA quality of the tissue block at this stage was based on the determination of the RNA Integrity of freshly collected tissue sections. FFPE tissue sections (10 μm) were performed and RNA purification was carried out using the RNAeasy FFPE extraction kit according to the manufacturer’s protocol (Qiagen, Hilden, Germany). Briefly, 2 sections of 10 µm thickness were cut for each block and sections were deparaffinized. Samples were digested using Proteinase K and DNase and RNAs were then purified on RNeasy MinElute spin columns from Qiagen. RNAs were recovered after elution in RNAse free water. The RNA integrity and quality were then first determined using the Nanodrop spectrophotometer to measure RNA concentration and the Agilent RNA 6000 nano kit bioanalyzer to evaluate the percentage of RNA fragments longer than 200 nucleotides (DV200 index) needed for the RNA Visium gene expression.

Five-micron FFPE blocs and HPS stained sections were first analyzed and validated for the presence of NCG structures and muscle tissue. Twenty-five mm^2^ areas were selected in tissue sections for each patient using the Case Viewer software (3DHISTECH, Budapest, Hungary, v2.3). Sections were then floated in the microtome bath at 42 °C until samples were flat without any wrinkles. Samples were then transferred on a Visium slide and placed in a drying rack for 3 h at 42 °C. Slides were then successively immerged in xylene and ethanol baths and were incubated at 37 °C in a thermocycler for 15 min for deparaffinization. Immediately after incubation, the slide was stained successively with hematoxylin and alcoholic eosin solution. Imaging was performed according to 10x Genomics recommendations on Pannoramic Scanner (3DHISTECH, Budapest, Hungary, v2.1.2104194).

In order to analyze expressions in FFPE biopsies, tissues were de-crosslinked using Hydrochloric Acid Solution (0.1 N), then with Tris EDTA buffer at 70 °C for 60 min. Human probes hybridization, ligation, probes release, and extension steps were realized following manufacturer instructions (10x Genomics). Illumina indexes were added to the libraries and sequenced on Novaseq 6000 Illumina sequencer (San Diego, CA, USA), targeting 50,000 reads/spot.

### 2.6. Statistical Analysis and Software

The Space Ranger software (10x Genomics, v1.3.0) was used to process the raw data. Spatial transcriptomics analyses were carried out by following two complementary approaches: (1) an unbiased approach, which uses the transcriptomic information of each spot and determines the transcriptomic proximity between all spots, with R (v4.1.0) package Seurat (v4.2.0), Spata2 (v0.1.0), and Monocle3 (v1.2.9); and (2) a histologically guided approach, which delineates the area of interest based on histological consideration only, using Loupe browser software (v6.2.0). Both approaches allow the exploration of the spatial transcriptomic datasets according to the scientific questions, i.e., the repartition of the spot with similar transcriptomic profiles vs. the transcriptomic differences between 2 histologically distinct areas.

All datasets were imported using SPATA and normalized using the log transformation method from the Seurat R package. Dimensionality reduction (Principal Component Analysis) was performed on the normalized data (RunPCA) on the 2000 most variable genes. The top 30 principal components were considered to generate a neighborhood graph (FindNeighbors). Data were then clustered using FindCluster with 0.4 and 0.6 as resolution parameters for INMG2 (patient #1) and INMG3 (patient #2), respectively. UMAPs were generated using the first 30 dimensions. Gene markers of each cluster were determined using the FindAllMarker function from the Seurat R package and differentially expressed genes between clusters were measured using MAST statistical tests (v1.20.0). Pathways corresponding to these transcriptomic differences were determined using the GO-BP (Gene Ontology Biological Processes) overrepresentation analysis (ClusterProfiler R package, v4.2.2).

## 3. Results

### 3.1. Cases Selection

Here, we report two patients identified for muscular sarcoidosis through the screening of the MYOLYON database. Patient #1 is a 84-year-old woman presenting with a bilateral proximal motor deficit and myalgia involving the lower limbs (MRC = three in psoas muscle). A diagnosis of muscular sarcoidosis was decided based on a ^18^F-fluoro-deoxyglucose (FDG) positron emission tomography (PET) scan (PET-scan) and muscular biopsy. The PET-scan revealed a diffuse muscular hypermetabolism, predominantly involving lower limbs and a mediastinum lymph node (Figure 1A,B). The hematoxylin–phloxine–saffron (HPS) staining of quadriceps muscle biopsies revealed the presence of massive predominant muscular granulomas (>1 mm; see asterisks in Figure 1G) distributed in the perimysium and associated with muscular atrophy, necrosis, and fibrosis. The electroneuromyography (ENMG) proved unremarkable.

The patient initially improved under 1 mg/kg/day corticosteroid therapy but relapsed later. As a second line therapy, she was treated using corticosteroid and methotrexate (15 mg/week), which were changed for corticosteroid and hydroxychloroquine because of poor hematological tolerance of methotrexate. At last follow-up, the PET-scan was normalized with an improvement in muscular symptoms.

Patient #2 is a 43-year-old woman presenting with malignant hypercalcemia without objective muscle deficit at muscular testing. A biological assessment found isolated hypercalcemia at 4 mmol/L. A PET-scan revealed a diffuse muscular hypermetabolism predominantly involving the lower limbs (Figure 1C–F), and a CT-scan revealed mediastinum adenopathy. The HPS staining of quadriceps muscular biopsy revealed a huge number of smaller granulomas (<500 μm; see black arrows) more widely distributed throughout the biopsy section and associated with muscular atrophy, necrosis, and fibrosis (Figure 1H). The ENMG was unremarkable.

The patient was improved but relapsed under 1 mg/kg/day corticosteroid therapy. As a second line therapy, she was treated with corticosteroid and methotrexate 10 mg/week allowing for a good control of the disease.

The granuloma structure characterization found in both patient biopsies was carried out via immunohistochemistry (IHC) using anti-CD68, anti-CD163 and anti-CD3, or anti-CD8 antibodies as markers of the monocyte/macrophage and lymphocyte lineages, respectively. Our data revealed that granulomas were mainly composed of CD68 and, to a lesser extent, of CD163 monocytes/macrophages and of CD3 T-lymphocytes. Only very few dispersed CD8-positive T-cytotoxic lymphocytes were observed in both biopsies, correlating well with previously published data [2,17,18] (Figure 2).

Considering that muscular sarcoidosis is associated with a wide range of phenotypes potentially linked to the size and the number of granuloma infiltrates, these two patients with very different granulomatous infiltrations in terms of pathology were both included in our study to identify possible transcriptomic signatures characterizing granuloma-muscle interaction.

### 3.2. Spatial Transcriptomics Analysis Identifies Five Different Clusters in Muscle Tissue Sections

The communications and interactions between granulomas and the surrounding muscle tissue, and the impact of one on the other, have not been investigated in depth. To address this important question, we employed the spatial transcriptomics experimental approach that allowed us to assess the expression of several thousand genes with subcellular and spatial resolution across millions of cells in the tissue sample. For this purpose, we used the 10x genomics Visium platform, a new barcoding-based spatial transcriptomics technology, to generate spatial maps of gene expression within granuloma and muscle tissue. This technology (summarized in Figure 3) proved to be a promising approach for exploring the dynamic and heterogeneous nature of granuloma infiltration and expansion, particularly in retrospective studies.

Overall, 16,848 genes were detected for the Patient #1 biopsy across 2723 capture spots analyzed with a 273 × 10^6^ depth of read and a median UMI (Unique Molecular Identifier) per spot of 12,385. For the patient #2 biopsy, 15,979 genes were detected across 2420 spots analyzed with a 222 × 10^6^ depth of read and a median UMI per spot at 4744.

After data normalization, an unsupervised data-driven approach generating unbiased predicted clusters based on transcript abundance distribution between spots, identified five major clusters (Figure 4A). These clusters encompassed the peri-granuloma tissue (cluster #0; PGT), the granuloma structures themselves (cluster #1; GS), and extended into three successive muscle areas corresponding to the proximal granuloma muscle area (cluster #2; PM), the intermediate area (cluster #3; IM), and the most distal muscle area (cluster #4; DM; Figure 4A, middle panel). Notably, these clusters, characterized by their transcriptomic distinctions, as depicted in the UMAP representation, closely mirror the histological segregation (Figure 4A, right panel). Interestingly, the PGT cluster tightly segregates with the granuloma structure while muscle PM, IM, and DM clusters were notably distinct from GS and PGT and gathered together. A similar clustering approach was also applied to the Patient #2 biopsy sample (Figure 5A).

To evaluate the strength of the data generated using Visium technology, transcriptomic data were first challenged with those issued from IHC that were previously conducted to characterize the cellular composition of the granuloma (Figure 2). The data presented in Appendix A revealed, although with more sensitivity, a clear correlation, between the spatial RNA gene expression profile of *CD68* and *CD8A* with the immunostaining data in both patient biopsies. Interestingly, this set of data also highlighted that low levels of mRNA expression were also correlated with a low immunostaining signal (Appendix A). Taken together, these data confirmed that our approach appeared robust enough to identify transcriptomic expression signatures across different tissue areas.

### 3.3. Evaluation of the Granuloma Structure’s Impact on the Surrounding Muscle Tissue through a Spatial Transcriptomics Approach

To determine if the muscle tissue cellular transcriptomic profile varies according to its proximity to the granuloma structure, we conducted a pseudotime analysis using Monocle algorithm, considering GS spots as starting points and the most distal muscle tissue (DM) as healthy since histology and transcriptome were normal in this region. In this analysis, pseudotime served as an indicator of transcriptomic distance from the healthiest part of the muscle and was used to order the spots along this gradient. The heatmap in Figure 4B (right panel) revealed that the transcriptomic profile of each spot evolved along the distance to the closest granuloma structures, even within cluster #2-PM which represents a transition between the GS/PGT area and the muscle tissue. The heatmap also highlighted that the cluster #0-PGT exhibited a unique transcriptomic profile, highly contrasting with the granuloma and the muscular areas despite its spatial proximity to both tissues (Figure 4B). Similar findings were observed in the second patient’s biopsy, although the contrasts between the defined clusters were less pronounced due to the granuloma size, their concentration, and their distribution across the tissue section (Figure 5B).

The upregulated genes identified in each cluster from both biopsies are presented in Figure 4B and Figure 5B (right panels) and in Appendix A. For Patient #1, the most upregulated gene in PGT were the *proteoglycan-4* (*PRG4*) or the *phospholipase A2 group IIA* (*PLA2G2A*) mRNAs, usually expressed by undifferentiated cells or fibroblasts [19] (Figure 4B,C and Appendix A). The GS cluster displayed a high expression of the myeloid markers of the monocyte–macrophages lineage, including the *Lysozyme C* gene (*LYS*) (Figure 4B,C). As expected, muscle markers like *DESMIN* (*DES*), *TITIN* (*TTN*) or the *Bridging Integrator 1* (*BIN1*) were expressed in PM, IM, and DM muscle clusters (Appendix A and Figure 4). Similar data were observed in the second patient’s biopsy (Appendix A and Figure 5).

To identify the biological processes involved in muscular sarcoidosis, upregulated genes (Appendix A) were selected in each cluster and enrichment analysis using the Gene Ontology (GO) pathway database was conducted to highlight potentially induced or impaired biological pathways in both patients. In the GS-cluster, GO Enrichment analysis predominantly revealed immune cell activation and cytokine production pathway signatures, suggesting an important network of paracrine communication. Notably, the analyses identified the activation of the Th1-CD4+ T-cell and CD68+ myeloid cells, as well as the upregulation of the expression of cytokines such as TNF-α, TGF-β, Interferon γ, IL-1/IL-2/IL-6/IL10, and IL-12 as significantly upregulated pathways (Figure 6A). Interestingly, enrichment analysis realized on the PGT-cluster highlighted the upregulated pathways related to collagen-containing extracellular matrix formation and organization as well as the activation of the TGF-β signaling, indicating the presence of an ongoing fibrotic process in the tissue surrounding the granuloma (Figure 6B). Applying a similar analytical approach to the second patient’s biopsy confirmed the findings observed for patient #1 (Figure 6C,D).

### 3.4. The Granuloma Structure Generates a Gradient of Transcriptomic Alterations in Muscle Fibers

To assess the impact of the granuloma structure on the neighboring skeletal muscle tissues, we compared the proximal muscle (PM) cluster to the granuloma with the most distal muscle clusters (DMs) in both patients. Transcriptomic differences between the PM and DM clusters were analyzed using the GO pathway database (Figure 6E,F and Appendix A). Signatures of inflammatory signaling pathways such as cellular response to interferon types I and II, TNF-α, and IL-1, as well as epithelial to mesenchymal cell transition or fibroblast proliferation pathways, were significantly increased in PM (Figure 6E and Appendix A (left)). This set of data was correlated with an upregulation of the downstream expression of *Serum amyloid A1* and *A2* (*SAA1* and *SAA2*; Appendix A and Figure 4B and Figure 5B, right panels) which is of interest since *SAA1* has been previously found to be induced by IL-6 or TNFα cytokine stimulation [20,21]. Remarkably, while inflammatory response pathways were upregulated in the PM-cluster, we concurrently observed a significant downregulation of skeletal muscle development, organization, and function pathways (Figure 6F and Appendix A (right)). Indeed, specific muscle gene markers, including *Myozenin 1* (*MYOZ1*), *Actinin 3* (*ACTN3*) or *Myosin light chain 11* (*MYLPF*) involved in muscle ultrastructure, were significantly downregulated in PM compared to the IM and DM-clusters (Figure 7A). This downregulation was present independently of muscle fiber types as defined by the expression markers such as *Myosin Light Chain 3* (*MYL3*), *Myosin Heavy Chain 2* (*MYH2*), *Myosin Heavy Chain 1* (*MYH1*) or *Tropomyosin1* (*TPM1*; Figure 7A and Appendix A). Interestingly, the expression of the downstream effectors of inflammatory cytokines like TGF-β or TNF-α and interferon-γ gradually declined with increasing distance from the granuloma; the expression of *Collagen 1A1* (*COL1A1*), *Endoglin* (*ENG*) and *Disabled homolog 2* (*DAB2*), *Interferon induced transmembrane protein 3* (*IFITM3*), *CC chemokine ligand 19* (*CCL19*), *Interferon regulatory factor 1* (*IRF1*), *SAA1*, and *SAA2* were all upregulated in the PM-cluster and progressively decreased through the IM and DM-clusters in both patients (Figure 7B,C).

The class-II major histocompatibility complex factor HLA-DRA (human leukocyte antigen DR isotype) is a downstream effector of interferon-γ and TNF-α stimulations [22,23] and has been recently implicated in granuloma formation in cutaneous sarcoidosis [24]. HLA-DRA is not well expressed in healthy skeletal muscle excepted in the endothelial cells of arterioles, venules, and capillaries [25]. However, in different muscular disorders, including polymyositis or dermatomyositis, HLA-DRA is upregulated in mononuclear inflammatory infiltrates and at the level of the sarcolemma and inside myofibers [25,26]. Strikingly, *HLA-DRA* expression was upregulated inside the granuloma clusters of both muscle biopsies (Figure 4B and Figure 5B, right panels). Interestingly, *HLA-DRA* was also upregulated in the PM-cluster and its expression gradually decreased when the distance to the granuloma increased (Figure 8A,B). IHC experiments confirmed the strong protein expression of HLA-DRA in the granuloma and in PM, at the sarcolemma, and inside the myofibers (Figure 8C,D).

Overall, the Visium transcriptomics approach enabled us to spatially visualize how granulomas impact the surrounding muscle tissues over short and long distances, revealing the establishment of an inflammatory and fibrotic gradient that disrupts skeletal muscle homeostasis in the proximity of the granuloma.

### 3.5. The Granuloma Structure Is Characterized by a Pro-Fibrotic Macrophage Signature

Recently, Coulis et al., through a combination of single-cell and spatial transcriptomics approaches, identified a predominant profibrotic macrophage cluster expressing the fibrotic markers *galectin3* (*Lgals3*), *Triggering Receptor Expressed on Myeloid Cells 2* (*Trem*)*2*, and *cathepsins -d, -l* and *-s* (*Cts-d, -l*, and *-s*) as well as *osteospontin* (*Spp1*) and *transmembrane glycoprotein nmb* (*Gpnmb*) in the dystrophic muscles of a Duchenne dystrophy myopathy (DMD) mdx mouse model and in human DMD muscles [15]. Interestingly, *Lgals3*, *spp1*, and *Gpnmb* were all previously involved in the fibrogenesis process [27,28,29,30] and Spp1 was found to promote the fibroblast expression of the *metalloproteinase 9* (*Mmp9*) and enhances TGF-β processing to induce collagen expression in skeletal muscle fibroblasts [31]. To determine if such profibrotic factors could be identified in granulomas, expression levels of *LGALS3*, *TREM2*, *GPNMB*, *SPP1*, and *CTS-D*, *-L*, and *-S* were evaluated in the patient #1 biopsy across the previously defined five clusters. The data in Figure 9 revealed a significant upregulation of these factors in GS and this correlated with the upregulation of extracellular matrix markers including *collagens* (*COL1A1*, *COL1A2*, *COL3A1*, *COL6A1*, and *COL6A3*) or *fibronectin* (*FN1*). Conversely, the human orthologs *folate receptor β* (*Folr2/FOLR2*), *Lymphatic Vessel Endothelial Hyaluronan Receptor 1* (*Lyve1/LYVE1*), growth arrest-specific gene 6 (*Gas6/GAS6*), *Fc Gamma Receptor And Transporter gene* (*Fcgrt/FCGRT*), *selenoprotein P* (*Sepp1/SELENOP*), *FXYD Domain Containing Ion Transport Regulator 2* (*Fxyd2/FXYD2*), *Metallothionein 1A* (*Mt1/MT1A*) or *Leukotriene C4 synthase* (*Ltc4s/LTC4S*), previously identified as signatures of skeletal muscle resident macrophages enriched in wild-type mouse muscles [15], appeared reduced (6/8) in the GS cluster but were upregulated in the muscle clusters (Figure 9).

Taken together, these findings suggest that granulomas exhibit a macrophage profibrotic signature.

## 4. Discussion

The cellular diversity within skeletal muscles plays a crucial role in muscle homeostasis and in the behavior of muscle stem cells and myogenic cells. Fibroblasts, immune cells, and endothelial cells, in the vicinity of muscle satellite cells and muscle fibers, communicate with each other to control muscle regeneration upon injury.

T-CD8+ lymphocytes, for example, are known to damage muscle fibers in dermatomyositis [32]. Macrophages are key orchestrators of inflammatory and regenerative processes after injury. Being either proinflammatory or anti-inflammatory, they have the dual ability to either inhibit or enhance muscle regeneration and they control the resolution of inflammation, a mandatory step for tissue repair [13]. However, although monocytes/macrophages are essential in tissue repair, in muscular degenerative pathologies such as DMD, their dysregulation was found to promote the development of muscle fibrosis [14,33].

Sarcoidosis is characterized by the presence of invading granulomas composed of immune cells such as macrophages and T-lymphocytes. Considering the particular case of muscular sarcoidosis, this study focused on the impact of these invading granulomas on the adjacent muscle tissue. For this purpose, we used a novel powerful next-generation molecular profiling strategy enabling the spatial visualization of gene expression to analyze cellular interactions and communications between granulomas and muscle fibers. This allowed us to show that sarcoidosis granulomas primarily consist of monocytes/macrophages and T-lymphocytes which produce a large array of cytokines potentially involved in paracrine communication. Spatial analyses revealed a proximo-distal gradient starting from the granuloma of TNFα, Interferons Ι, II, and TGF-β pathway activation in muscle cells, suggesting that the granulomas act as paracrine structures that send signals mediated by the release of cytokines/chemokines and/or extracellular vesicles carrying messengers. In lung sarcoidosis, TGF-β is known to participate to interstitial lung fibrosis [34]. In skeletal muscle, TGF-β is the major pathway that induces muscle atrophy and fibrosis in pathological situations, thereby inhibiting muscle regeneration [35]. TNF-α was previously found to induce fibrosis and different studies suggested that it can be mediated through the upregulation of TGF-β [36,37]. However, during the inflammatory phase after muscle injury, TNF-α released by infiltrating macrophages can also mitigate the fibrosis process by inducing the clearance of the fibro/adipogenic progenitors FAPs [38]. FAPs are essential for myogenic differentiation and regeneration. Nawaz et al. (2022) recently better characterized the crosstalk between CD206-positive macrophages and FAPs and showed that the depletion of CD206-macrophages promotes muscle regeneration through the activation of FAPs that secrete the promyogenic factor Follistatin [39]. The deletion of the FAP-specific *follistatin* gene delayed muscle regeneration and enhanced fibrosis. Mechanistically, the authors identified that the inhibition of FAPs activation by CD206-macrophages was mediated by the TGF-β signaling pathway. Interestingly, Prokop et al. (2011) identified that the CD206 marker was significantly upregulated in muscle biopsies of the muscle sarcoidosis group [40] and Preusse et al. identified that TGF-β co-localized with CD206 macrophages [8]. In this context, it is quite conceivable that communication between macrophages and FAPs in muscular sarcoidosis could be altered, thus leading to a profibrosing ground. Overall, these data confirm the high complexity of the sequential mechanisms driving the fibrosis pathway and the interconnections between cytokines pathways such as TNF-α and TGF-β.

Altogether, our results indicate that invading granulomas probably generate pro-inflammatory and pro-fibrotic signals which inhibit myogenesis and promote fibrosis, leading to clinical muscle atrophy and strength loss.

The pathophysiology of muscular sarcoidosis is poorly understood and only few studies in the literature have investigated the impact of granulomas on the muscular tissue. Preusse et al. studied granuloma formation in muscles through a comparison of T-helpers lymphocytes and macrophages’ fate in the case of muscular sarcoidosis and macrophagic myofasciitis [8]. In their study, they compared the RNA expression of a limited number of genes (16 genes involved in macrophage and T cell fate) using quantitative RT-PCR in laser-microdissected granulomas versus adjacent muscle layers. Their data revealed that the M2 polarization of CCL18-positive macrophages with TYROBP and TGF-β expression was crucial for granuloma formation. These results are consistent with our spatial transcriptomic data which emphasize the importance of TGF-β in muscular sarcoidosis.

Interestingly, Coulis et al. identified a profibrotic macrophage profile expressing *Lgals3*, *Trem2*, *Cts-d*, *-l*, and *-s* as well as *Spp1* and *Gpnmb* markers in mdx mouse dystrophic muscle [15]. Unbiased analyses carried out on Patient #1 revealed this signature concomitantly with fibrosis markers in our granuloma structures, which strongly suggests that granulomas are a source of profibrosing components that could potentially impact the surrounding muscle tissues.

We acknowledge the limitations of this study. First, only two patients were analyzed in this study and patient #2 displayed an unusual presentation for muscular sarcoidosis with a major and almost exclusive granulomatous muscular involvement (muscular biopsy, PET-scan) but without objective motor deficit. This unusual presentation has already been described in the literature as case-series or case report and was characterized as muscular sarcoidosis by authors who considered that muscular involvement was at the forefront of the pathology [41,42,43]. The size, the number, and the distribution of granulomas in patient #2 were different compared to those observed in the first patient, possibly explaining some signature nuances observed between the two biopsies. To circumvent these differences in granuloma distribution, a pseudotime analysis was performed with GS spots considered as starting points. The transcriptomic distance between each spot of the slide and the GS region was calculated. This arbitrary distance enabled us to well order spots on patient #1 but not on patient #2, due to the concentration distribution of granuloma across the tissue section. The second limitation is the resolution and the fine characterization of cells residing in the granuloma or in the muscle tissue. Indeed, the resolution of spatial transcriptomics analysis means that each spot encompasses several cells, thus we cannot exclude that transcriptomic findings could be biased in areas with the presence of infiltrating cells of different origins, such as, for example, the interface between the PGT and the PM areas. The best approach to avoid this pitfall could be mediated through complementary approaches, such as spatial transcriptomics combined with single nuclei RNAseq, in order to have a precise insight into the spatial and cellular transcriptomics information. However, this complementary approach was not possible due to the retrospective nature of our study. The precise evaluation of macrophage composition and abundance was not determined as previously performed by Coulis et al. [15]. However, in agreement with this study, the detection of signatures that may correspond to resident macrophages in muscle tissues, as well as profibrosing macrophage profiles in the granuloma during muscular sarcoidosis, could explain the heterogeneity between patients and their response to the different treatments engaged with.

## 5. Conclusions

To the best of our knowledge, this is the first study that investigates the pathophysiology of muscular sarcoidosis using the spatial transcriptomics approach. It shows that this strategy can be very useful in determining the cellular exchanges allowing the establishment, maintenance, and pathogenicity of granulomas on the surrounding tissue.

Our results tend to present this pathology as evolving towards a fibrosing process of the muscle which will no longer be accessible to treatment, perhaps explaining the evolution towards a state of disability of some patients. The challenge is to treat the patient quickly and strongly to avoid the evolution towards fibrosis. Clinicians should probably rapidly introduce long term immunosuppression in parallel with steroid use in the case of therapeutic failure to avoid fibrosis [44,45].

Therapeutic strategies targeting upregulated signaling pathways identified from our approach could be useful for muscular sarcoidosis treatment. Indeed, TNF-α inhibitors could decrease the inflammatory signaling pathway and thus could restore muscle homeostasis. These kind of treatments are known to be useful in cases of refractory sarcoidosis, but no evidence for their effectiveness in the particular case of muscular sarcoidosis have been previously described in the literature [46]. JAK-inhibitors could be an interesting therapeutic strategy via interferon-γ inhibition, as previously described by Damsky et al., in cutaneous sarcoidosis [24].

## Figures and Tables

**Figure 1 cells-12-02747-f001:**
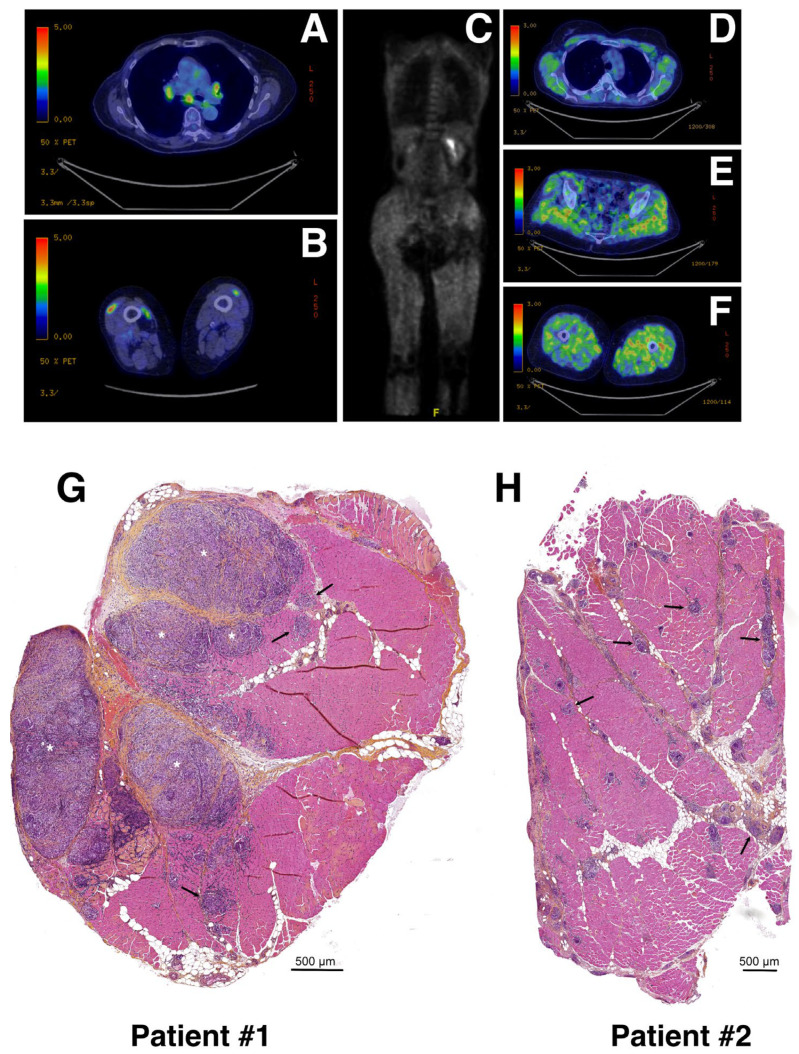
Radiological features and biopsies of two patients with muscular sarcoidosis. (**A**–**F**) A whole-body ^18^F-fluoro-deoxyglucose (FDG) positron emission tomography (PET) scan (PET-scan) showed a diffuse muscular hypermetabolism, predominantly involving the lower limbs and a mediastinum lymph node. Patient #1 (**A**,**B**) and patient #2 (**C**–**F**). (**G**,**H**) Hematoxylin–phloxine–saffron (HPS) staining of quadriceps muscle biopsies from patient #1 (**G**) and patient #2 (**H**). Asterisks and arrows indicate massive (>1 mm) and smaller (≤500 μm) granuloma, respectively. Scale bars: 500 μm.

**Figure 2 cells-12-02747-f002:**
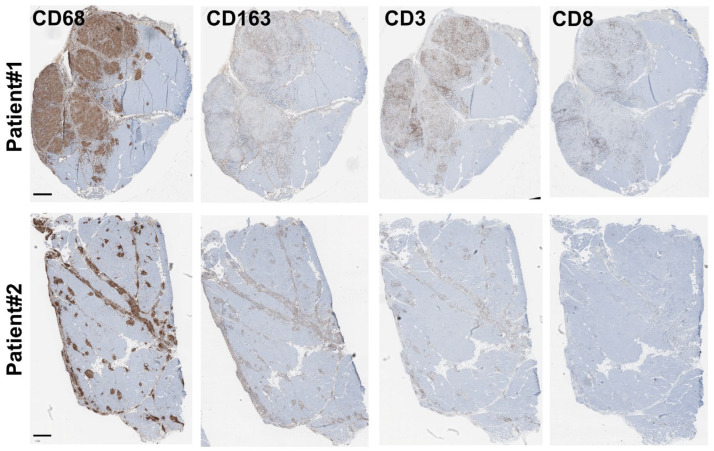
Granuloma characterization: immunohistochemistry characterization. Immunohistochemistry experiments were carried out on both biopsies (Patient #1: (**top**) and Patient #2: (**bottom**)) using antibodies directed against CD68, CD163, and CD3, or CD8 as markers of the monocyte/macrophage and lymphocyte lineages, respectively. Scale bar for all panels is 500 μm.

**Figure 3 cells-12-02747-f003:**
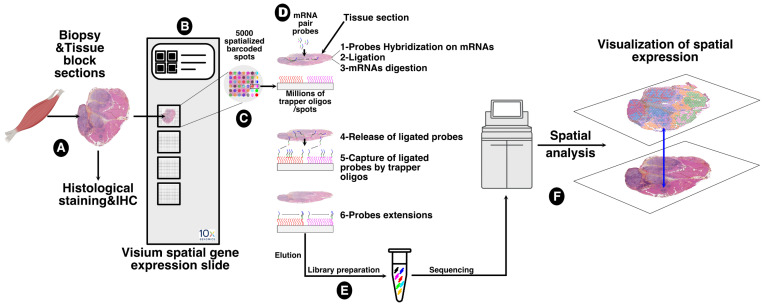
Sampling and workflow for the spatial transcriptomics approach. Tissue block sections from formalin-fixed paraffin embedding (FFPE) tissues of patients affected by muscular sarcoidosis containing granulomas (**A**) were placed on the capture area of a Visium spatial gene expression slide (10x genomics) (**B**) functionalized with 5000 spots, themselves containing millions of spatially assigned barcoded trapper oligonucleotides, ensuring that transcripts can be mapped to their original histological location (**C**). After FFPE tissue decrosslinking, specific pairs of probes were hybridized to tissue sections containing mRNA targets and ligated together. Ligated probes were released after tissue section permeabilization and mRNA digestion, and captured by the spatially assigned barcoded trapper oligonucleotides attached to the slide. An extension of the ligated probes, through reverse transcription, second strand synthesis, and amplification, was carried out to finalize the spatial assignment (**D**) and resulting DNA spatially barcoded probes were then eluted and used for DNA library preparation for Illumina high deep sequencing (**E**). Read sequencings were then processed and spatially resolved using the Seurat 4 algorithm. The data were then overlaid onto the acquired image of the tissue sections (**F**).

**Figure 4 cells-12-02747-f004:**
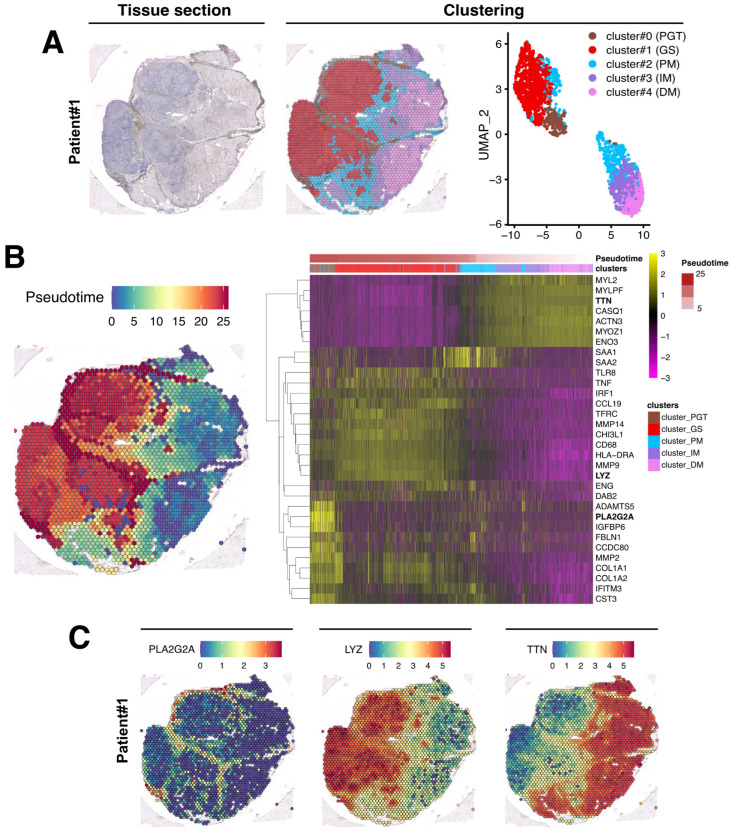
Spatial data analysis of muscular sarcoidosis—patient #1 biopsy. (**A**) Hematoxylin and eosin staining and projection of spot RNA clusters on the Visium slide as well as on UMAP. Visium array spots are color-coded based on cluster assignment of the integrated dataset. The perigranuloma tissue PGT cluster (brown); the granuloma structures GS cluster (red); the proximal muscle PM cluster (blue); the intermediate muscle IM cluster (purple) and the distal muscle DM cluster (pink). (**B**) Projection of the pseudotime on the Visium slide (**left panel**). Heatmap highlighting some marker genes of each cluster along the pseudotime, showing the gene expression modification along the gradient between granuloma and the “healthy-like” muscle tissue (**right panel**). (**C**) Expression of *PLA2G2A*, *LYZ*, and *TTN* on the Visium slide.

**Figure 5 cells-12-02747-f005:**
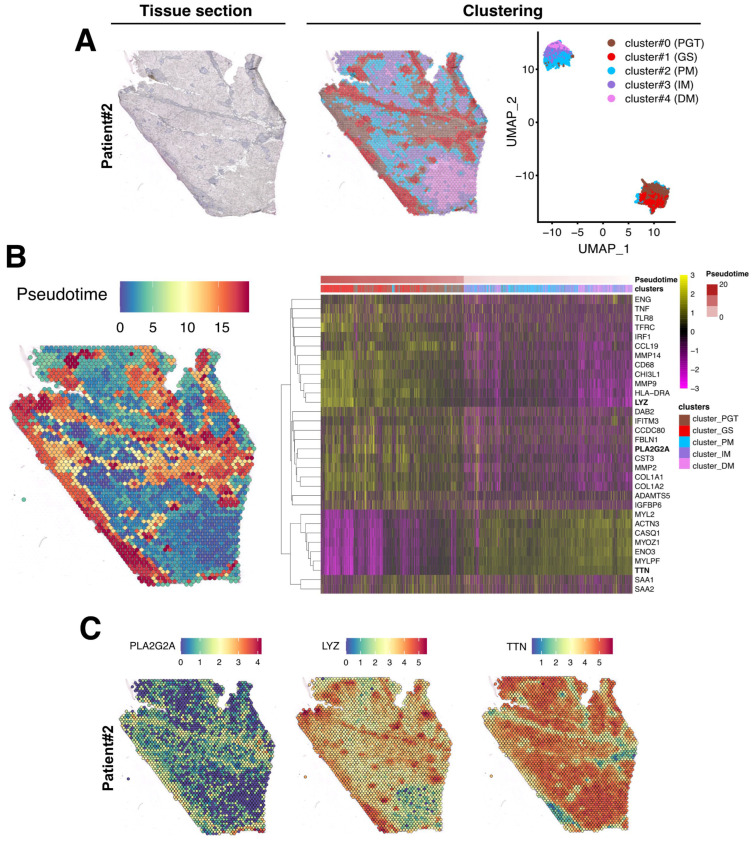
Spatial data analysis of muscular sarcoidosis—patient #2 biopsy. (**A**) Hematoxylin and eosin staining and projection of spot RNA clusters on the Visium slide as well as on UMAP. Visium array spots are color-coded based on the cluster assignment of the integrated dataset. Brown for the perigranuloma tissue PGT cluster; red for the granuloma structures GS cluster; blue for the proximal muscle PM cluster; purple for the intermediate muscle IM cluster; and pink for the distal muscle DM cluster. (**B**) Projection of the pseudotime on the Visium slide. Heatmap highlighting some marker genes of each cluster along the pseudotime, showing the gene expression modification along the gradient between the granuloma and the healthy muscle. (**C**) Expression of *PLA2G2A*, *LYZ*, and *TTN* on Visium slide.

**Figure 6 cells-12-02747-f006:**
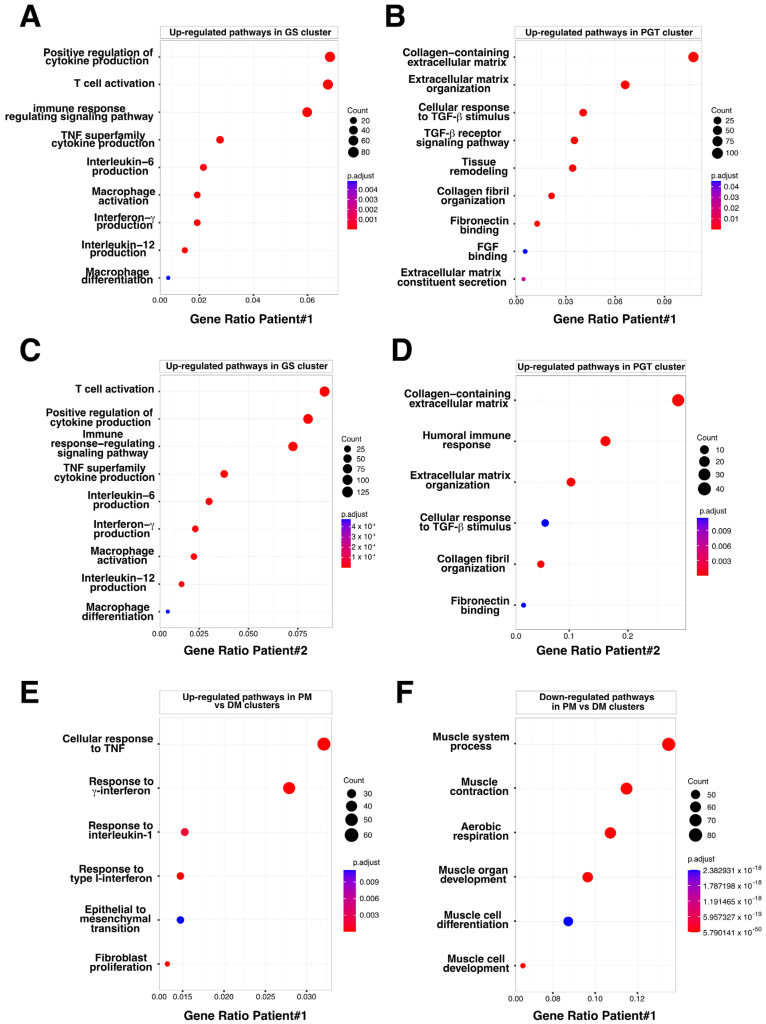
Bubble plot of GO-BP for Differentially Expressed Genes (DEG) in PGT, GS, and PM clusters. (**A**) Upregulated pathways identified in the granuloma structure GS cluster, Patient #1. (**B**) Upregulated pathways identified in the perigranuloma tissue PGT cluster, Patient #1. (**C**) Upregulated pathways identified in the granuloma structure GS cluster, Patient #2. (**D**) Upregulated pathways identified in the perigranuloma tissue PGT cluster, Patient #2. (**E**) Upregulated pathways identified in the proximal muscle PM cluster compared to the distal muscle DM cluster. (**F**) Downregulated pathways identified in the PM cluster compared to the distal muscle DM cluster. Gene ratio rich factor is the ratio of the DEG number to the total gene number in a certain pathway. The results are presented as bubble plots generated with the R package. FGF = Fibroblast Growth Factor; GS = Granuloma Structure; PGT = Peri Granuloma Tissue; PM = Proximal Muscle; TGF = Transforming Growth Factor; and TNF = Tumor Necrosis Factor.

**Figure 7 cells-12-02747-f007:**
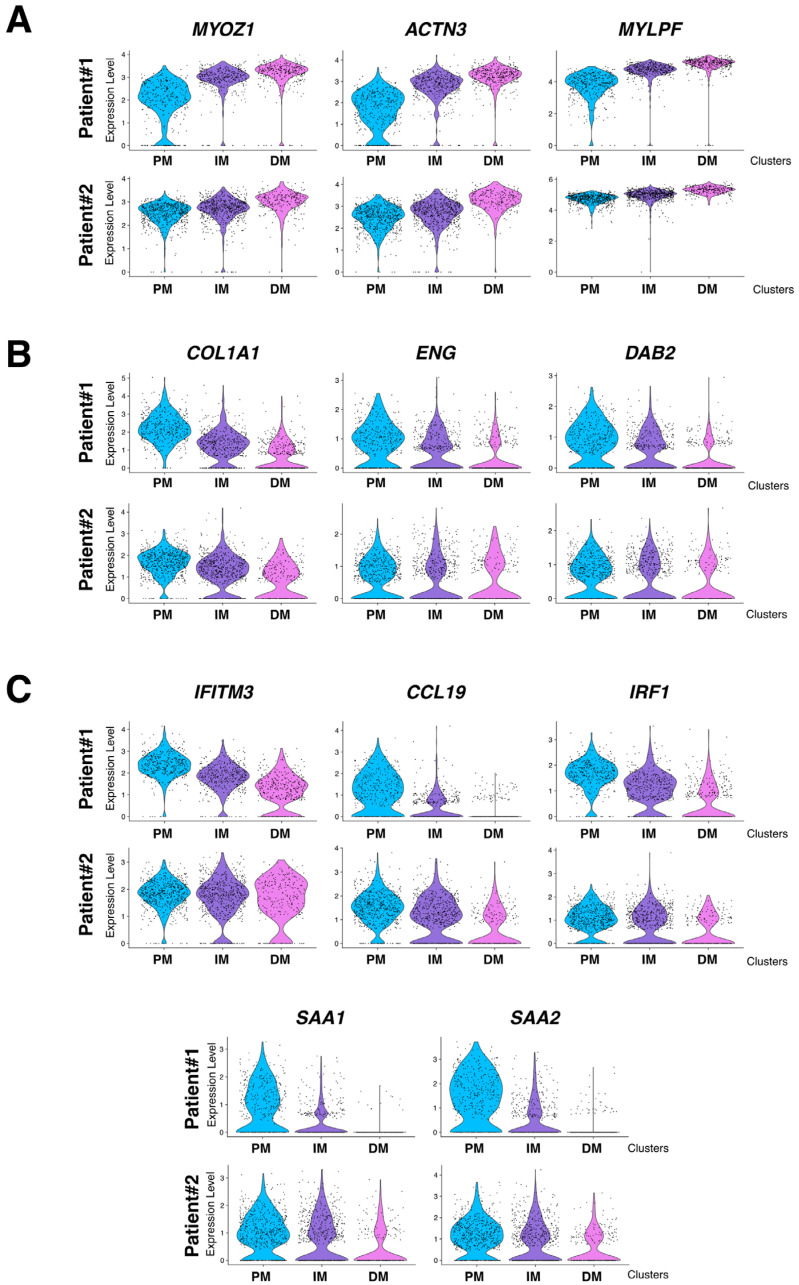
mRNA expression level of selected genes in PM, IM, and DM muscular clusters. (**A**) Expression level of selected muscle specific genes *MYOZ1* (*Myozenin 1*), *ACTN3* (*Actinin 3*), and *MYLPF* (*Myosin light chain 11*). (**B**) Expression level of TGF-β induced *COL1A1* (*Collagen 1A1*), *ENG2* (*Endoglin*), and *DAB2* (*Disabled homolog 2*) selected target genes. (**C**) Expression level of TNFα and Interferon-induced *IFITM3* (*Interferon induced transmembrane protein 3*), *CCL19* (*CC chemokine ligand 19*), *IRF1* (*Interferon regulatory factor 1*), *SAA1*, and *SAA2* selected target genes. Upper panel: Patient #1 and lower panel: Patient #2. PM = Proximal muscle; IM = Intermediate Muscle; and DM = Distal Muscle.

**Figure 8 cells-12-02747-f008:**
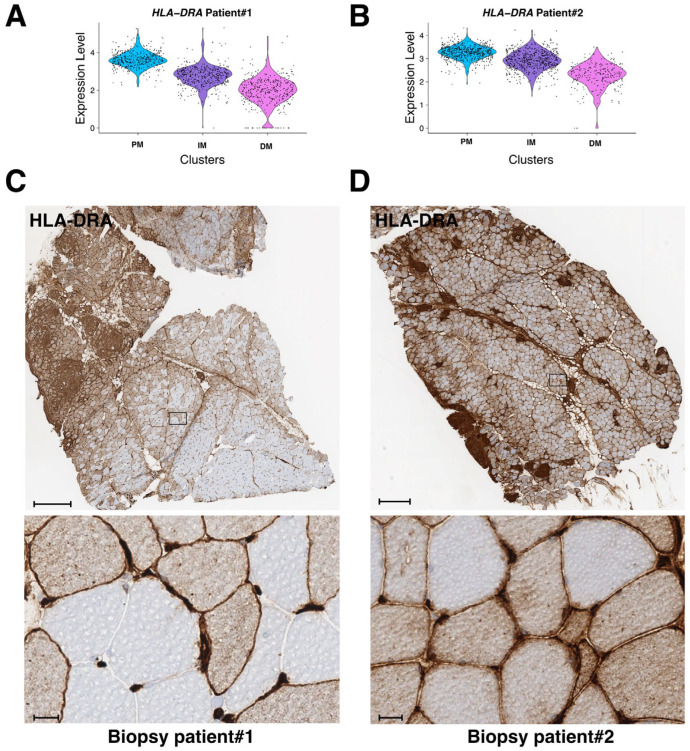
mRNA and protein expression level of HLA-DRA in PM, IM, and DM muscular clusters in both patients. (**A**,**B**) mRNA expression level of *HLA-DRA* in biopsy #1 (**A**) and patient biopsy #2 (**B**). (**C**,**D**) Immunohistochemistry analysis of HLA-DRA on frozen biopsies from patient #1 (**C**) and patient #2 (**D**). Lower panels correspond to higher magnification of selected regions (black rectangle in upper panels). Scale bars are 500 μm and 20 μm, respectively.

**Figure 9 cells-12-02747-f009:**
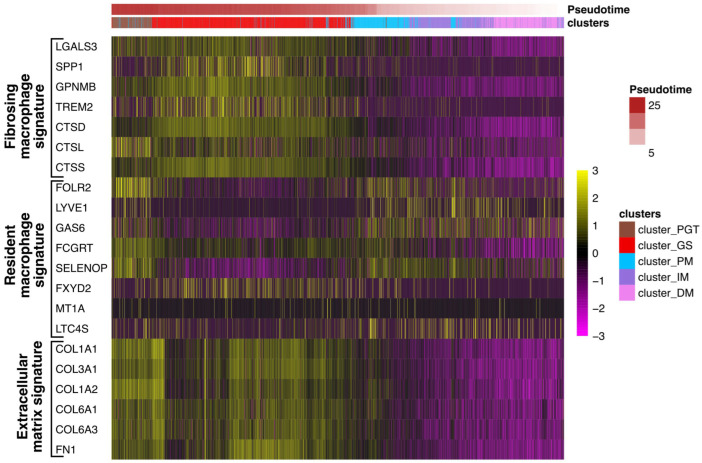
Profibrosing signature in the granuloma structure. Heatmap showing the expression of genes corresponding to signatures of resident macrophages (*FOLR2*, *LYVE1*, *GAS6*, *FCGRT*, *SELENOP*, *FXYD2 MT1A*, and *LTC4S*) identified in healthy muscles or *LGALS3*, *SPP1*, *GPNMB*, *TREM2*, *CTS-D*, *-L*, and *-S*, as previously described in profibrosing macrophages^15^, as well as markers of the extracellular matrix (*collagens*: *COL1A1*, *COL3A1*, *COL1A2*, *COL6A1*, *COL6A3*, and *fibronectin FN1*) in each cluster along the pseudotime analysis.

## Data Availability

Anonymized data not published within this article will be made available upon request to any qualified investigator. Raw data files issued from the Visium analyses are available at the Gene Expression Omnibus (*GEO*) repository database under accession number “GSE243291” (https://www.ncbi.nlm.nih.gov/geo/query/acc.cgi?acc=GSE243291; accessed on 25 November 2023) with GSM7782914 for INMG2-patient#1 and GSM7782915 for INMG3-patient#2.

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
