# Peer review of "Spatial Transcriptomics Reveals Signatures of Histopathological Changes in Muscular Sarcoidosis"

_cells, 2023, doi:10.3390/cells12232747_

Round 1
Reviewer 1 Report
Comments and Suggestions for Authors
This study provides a comprehensive analysis of spatial gene/protein expression in muscular sarcoidosis, molecularly linking invading granulomas and fibrosis. The results are well presented, and the discussion is in-depth. I would like to commend the authors for their efforts to study this rare disease with new spatial transcriptomics technologies. I have some comments that may help make the analysis more rigorous.
Major concerns:
1. Line 274: The authors stated that muscle PM, IM, and DM are notably distinct from GS and PGT, but the right panel of Figure 4A appears to show a group of PM near GS. Could the authors discuss this discrepancy? My guess is that because a spot contains multiple cells, these spots are a mixture of PM and GS. If this is the case, it is a good idea to exclude these spots in downstream analysis (such as for Figures 6, 7, 8) to avoid “contamination”. Figure 5 appears to have the same issue.
2. Is there a particular reason to have a unified trajectory/pseudotime for PGT/GS and PM/IM/DM? It is usually a good idea to infer a trajectory for each “partition” in Monocle because the trajectory linking two partitions can be arbitrary. This is evidenced by the fact that the ordering of cells by pseudotime is PGT, GS, PM/IM/DM in Figure 5, but GS/PGT, PM/IM/DM in Figure 6. I understand that a unified pseudotime for the whole slide may be appealing. In that case, it might be a good idea to explore the parameter setting of Monocle to achieve more consistent pseudotime ordering of cells across samples and mention the limitation of this approach (arbitrary ordering) in Discussion.
3. To corroborate Figure 2, the authors could consider plotting the genes for these markers detected by Visium in a supplementary figure. The protein and mRNA levels are known to not be highly correlated for some gen, so a general match is sufficient.
4. Are there notable differences in the samples from the two patients that are worth mentioning (given that they are indeed different as explained in Discussion)?
Minor concerns:
1. Cell types in the UMAP in Figure 5 can be similarly labeled as in Figure 4.
2. The font size of some texts in Figure 6 is too small. Also, the titles of the panels can be more informative (e.g., which patient; vs which cell types).
3. Figure 6E,F: which patient are these results for (and what about the other one)?
Typos:
1. Line 72: Unravel → Unraveling. Underline → underlie. (I’d suggest “Unraveling mechanisms underlying this disease”.)
2. Line 92: I believe 50 is the diameter (in microns), not the area (in squared microns).
3. Line 150: Qiagen?
4. Line 189: What does 1:30 neighbors mean? Which argument does it correspond to?
Author Response
Dear Reviewer 1,
Thank you very much for taking the time to review this manuscript and your help to let us the opportunity to improve it. Please find the detailed responses below and the corresponding revisions/corrections highlighted in red in the re-submitted file.
REVIEWER 1
This study provides a comprehensive analysis of spatial gene/protein expression in muscular sarcoidosis, molecularly linking invading granulomas and fibrosis. The results are well presented, and the discussion is in-depth. I would like to commend the authors for their efforts to study this rare disease with new spatial transcriptomics technologies. I have some comments that may help make the analysis more rigorous.
Reviewer Q1:
1. Line 274: The authors stated that muscle PM, IM, and DM are notably distinct from GS and PGT, but the right panel of Figure 4A appears to show a group of PM near GS. Could the authors discuss this discrepancy? My guess is that because a spot contains multiple cells, these spots are a mixture of PM and GS. If this is the case, it is a good idea to exclude these spots in downstream analysis (such as for Figures 6, 7, 8) to avoid “contamination”. Figure 5 appears to have the same issue.
Answer authors:
We understand that the UMAP representation vs clustering may appear confusing. These “contaminated” spots appear close to the GS and PGT clusters when considering the 2 first axes of the UMAP but their transcriptomes are closer to the other PM spots if we consider the first 30 components of the PCA like for the clustering. It is not a contamination but only a display problem of the UMAP. We have no statistical evidence to remove these spots only based on the 2 first UMAP axis and hide the fact that they are statistically closed to the other PM spots.
Reviewer Q2:
2. Is there a particular reason to have a unified trajectory/pseudotime for PGT/GS and PM/IM/DM? It is usually a good idea to infer a trajectory for each “partition” in Monocle because the trajectory linking two partitions can be arbitrary. This is evidenced by the fact that the ordering of cells by pseudotime is PGT, GS, PM/IM/DM in Figure 5, but GS/PGT, PM/IM/DM in Figure 6. I understand that a unified pseudotime for the whole slide may be appealing. In that case, it might be a good idea to explore the parameter setting of Monocle to achieve more consistent pseudotime ordering of cells across samples and mention the limitation of this approach (arbitrary ordering) in Discussion.
Answer authors :
The idea of pseudotime analysis was to order spots according to their transcriptomic distance to GS to be able to see a structuration of muscular tissue around GS, especially for Patient#2, which highlights a peculiar granuloma structure in patches. For this reason, we tested this analysis on the Patient#1 Visium, which shows a well-structured granuloma. The size of the granuloma compared to the Visium spots makes this ordering nice with PGT, GS, PM/IM/DM in Figure 5. In Patient#2, the granulomas are smaller compared to the Visium spots, which implies a mixture between GS and PGT in some spots and thus an organization of spots with
pseudotime analysis not as organized as for Patient#1, especially considering the fact that Monocle uses slightly different algorithms. We already try to change different parameter setting of Monocle but we did not achieve a more consistent pseudotime ordering of cells across samples.
As requested, we highlighted this point in the last paragraph of the discussion section page 17 lines 537-542 and 544 and 546.
Reviewer Q3 :
3. To corroborate Figure 2, the authors could consider plotting the genes for these markers detected by Visium in a supplementary figure. The protein and mRNA levels are known to not be highly correlated for some gen, so a general match is sufficient.
Answer authors:
This is an interesting question indeed that we investigated when our study was initiated. Briefly, one question we asked was to determine if the distribution and the protein expression level could be correlated with that of mRNAs. In this context we investigated two opposite expression levels (high and low) by focusing our attention on CD68 and CD8 respectively. Our data revealed a clear correlation between the protein expression level and their mRNA levels although the sensitivity brought by the spatial transcriptomics was higher compared to the IHC. This set of data was now included in the revised manuscript. A paragraph was added page 7 lines 277-285 and linked to a new supplementary Figure S1. The supplementary legend of Figure S1 is now depicted page 18 lines 579-583.
In consequence Figure S1 from the initial manuscript version became Figure S3 (see page 17 line 588).
Reviewer Q4:
4. Are there notable differences in the samples from the two patients that are worth mentioning (given that they are indeed different as explained in Discussion)?
Answer authors :
Despite the difference in granuloma structuration in patient#1 and #2, the muscle areas show similar transcriptomic pattern for both patients. This is now presented in Figure S2A,B page 12. The legend for Figure S2 is depicted page 18 lines 583-588.
Reviewer Minor concerns:
1. Cell types in the UMAP in Figure 5 can be similarly labeled as in Figure 4.
Answer authors:
We modified, as suggested, the Figure 5 to standardize with the Figure 4.
2. The font size of some texts in Figure 6 is too small. Also, the titles of the panels can be more informative (e.g., which patient; vs which cell types).
Answer authors:
We agree with Reviewer 1 and standardized the font size in this Figure 6. We also added the patient identification in the different panels and the titles are now more informatives.
3. Figure 6E,F: which patient are these results for (and what about the other one)?
Answer authors:
The data presented in Figure 6 E,F correspond to the Patient#1. We now indicated this information in the Figure 6 panels E and F, as well as we added this precision in the legend of the Figure. As mentioned above, the data for the patient#2 are now presented in Figure S2A,B.
Reviewer Typos:
1. Line 72: Unravel → Unraveling. Underline → underlie. (I’d suggest “Unraveling mechanisms underlying this disease”.)
Answer authors:
We modified as suggested the sentence page 2 in Line 73.
2. Line 92: I believe 50 is the diameter (in microns), not the area (in squared microns).
Answer authors:
We corrected this error and have deleted the “2” and precised 55 μm and not 50 page 2 line 93.
3. Line 150: Qiagen?
Answer authors:
We corrected Quiagen in Qiagen page 4 lines 151 and 154.
4. Line 189: What does 1:30 neighbors mean? Which argument does it correspond to? “The top 50 principal components were considered to generate a neighborhood graph using 1:30 neighbors (FindNeighbors).”
Answer authors:
We thank Reviewer 1 to have seen this error. Indeed, we used the top 30 principal components and not the top 50. This was corrected page 4 line 189.
Note that all figures (modified and unmodified) were now uploaded at a higher resolution.
Remark:
The Revised figures are not embedded with the CELLS formatted manuscript
We hope that the revised version of this manuscript and our responses to your comments will convince you and the Editor that our work is now suitable for publication in CELLS.

Reviewer 2 Report
Comments and Suggestions for Authors
Cudos for this very nice manuscript. The authors present spatial transcriptomics data on muscular sarcoidosis. The method is of clear novelty and of direct interest for the scientific community. I encourage publication.
Some limitations should be addressed beforehand
- One major limitation is that "only" two patients were assessed. Surely, spatial transcriptomics is expensive and it is a rare disease. Still, this limitation should be mentioned in the abstract and discussion.
- Fig. 1G is rather difficult to see. Please provide higher resolution for this image.
- Fig. 6: Please provide a homogenous typesetting. Please also assess the GO-BP dataset and wikipathways and add these enrichments as supplement. It is also unclear if this is Gene Set Enrichment or Overrepresentation analysis. GSEA would be advised as opposed to overrepresentation. It is advised to use ClusterProfiler as opposed to EnrichR and combine the "up" and "down" regulation into one plot for a better overview. For example as Ridge Plot.
- Please provide abbreviations for the figure legends.
- The authors might choose to highlight the value of single nuclei and transcriptomic analysis for the study of skeletal muscle in the discussion. These methods are likely to change our understanding of these pathologies and usage is encouraged, from my point of view.
- The authors should also highlight the role for fibro-adipogenic precursors (FAPs) for sarcoidosis and the instigation of fibrotic remodelling in the discussion. To keep the discussion concise, maybe drop the paragraph starting with "T-CD8+" (line 453 to 460).
Comments on the Quality of English LanguageThe english is fine from my point of view.
Author Response
Dear Reviewer 2,
Thank you very much for taking the time to review this manuscript and your help to let us the opportunity to improve it. Please find the detailed responses below and the corresponding revisions/corrections highlighted in red in the re-submitted file.
REVIEWER 2
Cudos for this very nice manuscript. The authors present spatial transcriptomics data on muscular sarcoidosis. The method is of clear novelty and of direct interest for the scientific community. I encourage publication.
Some limitations should be addressed beforehand
Reviewer Q1:
- One major limitation is that "only" two patients were assessed. Surely, spatial transcriptomics is expensive and it is a rare disease. Still, this limitation should be mentioned in the abstract and discussion.
Answer authors:
We agree with Reviewer 2 and, as suggested, we added now in the abstract that this study was conducted on two patients affected by muscular sarcoidosis page 1, line 31.
The notion of limitation of this study due to the low number of patients was previously described in the discussion section page 17 line 529-532.
Reviewer Q2:
- Fig. 1G is rather difficult to see. Please provide higher resolution for this image.
Answer authors:
We enlarged the panel 1G and uploaded this Figure 1 with a higher resolution.
Reviewer Q3:
- Fig. 6: Please provide a homogenous typesetting. Please also assess the GO-BP dataset and wikipathways and add these enrichments as supplement. It is also unclear if this is Gene Set Enrichment or Overrepresentation analysis. GSEA would be advised as opposed to overrepresentation. It is advised to use ClusterProfiler as opposed to EnrichR and combine the "up" and "down" regulation into one plot for a better overview. For example as Ridge Plot.
Answer authors:
We now increased and standardized the font size in this Figure 6.
We made a mistake in the Figure 6 legend, as we assessed GO-BP dataset using overrepresentation analysis instead of Scatterplot of Enriched KEGG pathways. Sorry for this error. This is now corrected in the new version of the Figure 6 legend page 18 line 583. Regarding the analysis on wikipathways data base, due to the short window time we had for the revision (only 5 days) the generated results obtained should have required more investigation time to identify if the revealed pathways are connected (based on the list of genes) to our study and pathology. In this context we only present the results from the GO-BP analyses. For Visium as for scRNAseq, the matrix has a lot of 0 counts which makes it difficult to applied GSEA based on gene ranking. In this context, we applied UCell, similar to ssGSEA.
We provid here (only for reviewer; see at the end of this document) the RidgePlot representation for a small set of the GO-BP pathways for patient#1. Note that similar results to those previously presented in Figure 6E,F were obtained. Due to the important volume of data generated by this approach, we decided not to show this representation in the manuscript and to only present the dot plot global representation.
Reviewer Q4:
- Please provide abbreviations for the figure legends.
Answer authors:
We know provided abbreviations in the different figure legends when necessary.
Reviewer Q5:
- The authors might choose to highlight the value of single nuclei and transcriptomic analysis for the study of skeletal muscle in the discussion. These methods are likely to change our understanding of these pathologies and usage is encouraged, from my point of view.
Answer authors:
We agree with Reviewer 2 that the combination of single nuclei associated with spatial transcriptomics is the best way to deeply characterized the different processes involved in this pathology. That is the reason why we indicated in the discussion section of the initial version that the best approach should be the single nuclei + spatial transcriptomics approaches page 17 lines 546-550. Unfortunately, due to the retrospective nature of our study , single nuclei transcriptomics approach could not be carried out.
Reviewer Q5:
- The authors should also highlight the role for fibro-adipogenic precursors (FAPs) for sarcoidosis and the instigation of fibrotic remodelling in the discussion. To keep the discussion concise, maybe drop the paragraph starting with "T-CD8+" (line 453 to 460).
Answer authors:
We thank Reviewer 2 and highlighted now the role of fibro-adipo progenitors in the discussion section (page 16 lines 495-507) on the crosstalk between macrophages and FAPs and the potential impact on the fibrotic remodelling process. Two additional references were added in this paragraph (references 39 and 40).
We chose to maintain the paragraph page 16 lines 469 to 476 because, from our point of view, it allows to introduce the notion that immune cells like lymphocytes and macrophages are essential modulators of the muscle regeneration process. Furthermore, it makes the transition with the next paragraph that is dedicated to the granuloma composition and the impact of granuloma immune cells on the surrounding muscle tissues.
Note that all figures (modified and unmodified) were now uploaded with a higher resolution.
Remark:
The Revised figures are not embedded with the CELLS formatted manuscript
We hope that the revised version of this manuscript and our responses to your comments will convince you and the Editor that our work is now suitable for publication in CELLS.
Note that a figure for reviewer only is associated with this associated pdf Response to the reviewer 2.
